# Basalt Fiber Composites with Reduced Thermal Expansion for Additive Manufacturing

**DOI:** 10.3390/polym14153216

**Published:** 2022-08-08

**Authors:** Daniel Moreno-Sanchez, Alberto Sanz de León, Daniel Moreno Nieto, Francisco J. Delgado, Sergio I. Molina

**Affiliations:** 1Departamento de Ingeniería Mecánica y Diseño Industrial, Escuela Superior de Ingeniería, IMEYMAT, Universidad de Cádiz, Campus Río San Pedro s/n, 11510 Puerto Real, Spain; 2Departamento de Ciencia de los Materiales e I. M. y Q. I., Facultad de Ciencias, IMEYMAT, Universidad de Cádiz, Campus Río San Pedro s/n, 11510 Puerto Real, Spain

**Keywords:** fiber-reinforced composites, basalt fiber, ASA, additive manufacturing, fused filament fabrication, mechanical properties, coefficient of thermal expansion, design, warping

## Abstract

Fused filament fabrication (FFF) is gaining attention as an efficient way to create parts and replacements on demand using thermoplastics. This technology requires the development of new materials with a reliable printability that satisfies the requirement of final parts. In this context, a series of composites based on acrylonitrile styrene acrylate (ASA) reinforced with basalt fiber (BF) are reported in this work. First, several surface modification treatments are applied onto the BF to increase their compatibility with the ASA matrix. Then, once the best treatment is identified, the mechanical properties, coefficient of thermal expansion (CTE) and warping distortion of the different specimens designed and prepared by FFF are studied. It was found that the silanized BF is appropriate for an adequate printing, obtaining composites with higher stiffness, tensile strength, low CTE and a significant reduction in part distortion. These composites are of potential interest in the design and manufacturing of final products by FFF, as they show much lower CTE values than pure ASA, which is essential to successfully fabricate large objects using this technique.

## 1. Introduction

Additive manufacturing (AM), also known as 3D printing, comprises a set of technologies that makes the manufacturing of parts directly from a digital model possible, generally in a layer-by-layer approach [1,2,3]. Among others, the polymeric material extrusion (PME) technologies are the most implemented in industrial sectors due to their cost efficient and easily scalable alternatives. In this field, fused filament fabrication (FFF, also known as fused deposition modeling, FDM) uses a filament as feedstock for the manufacturing of the parts, while fused granulated fabrication (FGF) uses pellets. As it happens with traditional manufacturing processes of thermoplastics such as injection molding and extrusion, AM processes requires the development of tailored materials. For PME, the material must ideally have a low coefficient of thermal expansion (CTE), an adequate melt flow index and good mechanical properties [4,5,6], so that both of the final requirements of the parts and the manufacturing process conditions are satisfied.

The use of amorphous thermoplastics and short fibers as reinforcements is well studied as a way to reduce the thermomechanical distortions that provoke cracking and warping issues in parts manufactured by AM [5,7,8]. Materials with a low CTE make the design and manufacture of parts that requires a high degree of dimensional stability possible, such as autoclave tooling by PME [7]. Loading polymers with fibers reduces the CTE and increases their stiffness, showing a better mechanical behavior against the internal tensions of the deposited material as it cools down, while it also provides a better heat dissipation and a reduction in distortion parts like warping. The most studied fibers for this purpose are carbon fibers (CF) and glass fibers (GF) [7,9,10,11,12]. 

Basalt fibers (BF) are gaining a lot of attention as a reinforcing material for polymers because of its properties and its natural origin. From volcanic rocks (composed mainly from plagioclase, pyroxene and olivine), BF has interesting properties such as a mechanical performance between GF and CF, high thermal resistance and insulation, intrinsic fireproof abilities and abrasion resistance at a reduced cost (eight times lower than CF) [13,14,15,16]. The use of BF in composites has been addressed in thermoplastic materials as well as in thermoset resins [17,18,19,20]. For example, thermosetting resins (mainly epoxy) are typically reinforced with continuous fibers in structural and civil applications [14,21]. Thanks to the high thermal resistance of BF, they keep their properties even at high temperatures [22,23]. In thermoplastic polymers, the influence of the mechanical response of milled BFs embedded in an ABS matrix was reported, obtaining the best results when using 5% of molten BF with ABS [24]. Other authors have combined BF composites with polyurethane [25] or polylactic acid (PLA) in the development or a biocompatible reinforced composites [26]. Some studies have already been conducted using BFs in AM technologies, employing ABS or PLA as polymeric matrix by FFF [27,28,29]. However, there is still a wide field ahead to further investigate these materials.

The good integration of the fibers (either CF, GF or BF) in the polymeric matrix is a key factor in order to exploit their good mechanical properties. In this regard, the surface functionalization of the fibers has been performed via chemical modification with maleic anhydride [20,25,30], silane coupling agents [31,32,33,34] or with inorganic acid and bases [35,36,37]. Plasma treatment has also been used for surface functionalization [38]. The use of these strategies has reported improvements in the mechanical properties of the composites. For instance, Taylor et al. used atmospheric plasma to increase the adhesion of BF to a PP matrix processed by FFF and increasing its flexural modulus by 12% [39].

In this work, a series of composites suitable for injection molding and FFF using BF as a filler using acrylonitrile styrene acrylate (ASA) as a polymeric matrix have been developed. ASA is an amorphous polymer, with well-balanced mechanical properties and an enhanced resistance to ultraviolet (UV) radiation, compared to ABS [40,41,42]. For this reason, ASA is preferred in the industrial sector for the manufacturing of parts and objects for outdoor applications. We have previously evidenced that the suitability of ASA as a polymer matrix for composites reinforced with CF in AM technologies [43]. However, to the best of our knowledge, no research has been conducted combining ASA and BF in the development of novel composites, neither for traditional nor for AM technologies.

For this reason, different parameters such as filler content and different functionalization strategies are explored in this study. The influence of these parameters in mechanical properties have been studied, as well as analyzed by spectroscopic and microscopy techniques. Finally, the influence of these fibers in the CTE of the composites is also assessed, proving that the use of short BF can significantly reduce this value, as well as affect the reduction in warpage. These improvements make these new materials rather interesting for AM applications, especially in those where large parts and objects are required. The use of ASA reinforced with BF will enable the production of manufacturing parts with higher mechanical resistance, especially for outdoors applications.

## 2. Materials and Methods

### 2.1. Materials

ASA LI 941 NC was purchased from LG Chem Ltd. (Seoul, Korea). Chopped basalt fibers (BF) of 0.16 mm length and 13 ± 1 µm diameter were purchased from Deutsche Basalt Faser GmbH (Sangerhausen, Germany). The BF composition, according to the supplier, is presented in Table 1. Sulfuric acid (H_2_SO_4_) and sodium hydroxide (NaOH) was purchased from Scharlab (Barcelona, Spain). 3-Aminopropyltriethoxysilane (APTES) was purchased from Alfa-Aesar (Kandel, Germany).

### 2.2. Surface Modification of the Basalt Fibers

The BFs were first calcined in a Mufla B180 ^®^ oven, Nabertherm GmbH (Lilienthal, Germany) for 1 h at 500 °C to remove any prior sizing by the manufacturer or any impurities of the BFs. After calcination, the BFs were cleaned several times with distilled water and dried for 2 h at 120 °C.

A treatment in either acidic or basic conditions was done by immersing the calcined BFs (BF-c) in a 1 M H_2_SO_4_ or 0.37 M NaOH solution for 45 min under magnetic stirring at 40 °C. A silanization sizing was done using an APTES solution in distilled water with a proportion of 1:18, under magnetic stirring overnight. In all cases, at least 50 g of calcined BFs (BF-c) were used in each surface modification treatment and the BFs were washed afterwards several times with distilled water, until the pH of the washing solution was neutral. Then, the treated fibers were dried in the oven for 2 h at 120 °C. Hereinafter, the BFs treated with acidic conditions, basic conditions and silanized will be referred as BF-a, BF-b and BF-s, respectively.

### 2.3. Manufacturing of the ASA-BF Composites

All the raw materials (ASA and BFs) and their composites were dried for at least 4 h at 80 °C to remove any residual moisture. Then, approximately 100 g of neat ASA and ASA with 5–10 wt% of the treated BFs (BF-c, BF-a, BF-b, BF-s) were processed in a single-screw Noztek Pro laboratory extruder (Noztek, Shoreham, UK, L/D 26:14 cm, 60 rpm). A temperature of 240 °C was used for ASA and 260 °C for all composites. In all cases, a continuous filament with a diameter of 1.75 mm was produced.

Part of these filaments was cut again in small pieces and used as feedstock for injection molding (IM). At least 5 normalized tensile specimens (dog-bone type 1BA according UNE EN ISO 527) of the different materials were injected in a Babyplast 10/12 P (Cronoplast SL, Barcelona, Spain). The temperature profile of the injector was 230–240–235 °C in the plasticization, chamber and nozzle areas, respectively.

Then, different objects were manufactured using a fused filament fabrication (FFF) 3D-printer Raise 3D Pro 2 (Impresoras 3D, Almería, Spain). The slicer software used was ideaMaker, using a linear infill of 100% with an overlap of 20% between beads to generate all the g-codes and a nozzle of Ø 0.4 mm. At least 5 normalized 1BA tensile testing specimens were printed using a linear infill at 0° (XY orientation) and vertical (XZ orientation), with a layer height of 0.2 mm (nomenclature according to AM standards [44]). Monolayers of 10 × 10 × 0.1 mm (length, width and height) were printed for chemical analysis via Fourier-Transformed Infrarred (FTIR) spectroscopy. Parallelepipeds of 20 × 5 × 5 mm and 5 × 5 × 20 mm (labelled as XY and XZ, respectively), with a layer height of 0.2 mm were printed to evaluate their coefficient of thermal expansion (CTE). V hollow-shaped parallelepipeds were printed to evaluate the warpage behavior. The shape and dimensions of those specimens are displayed in Figure 1. The objects were printed with a layer height of 0.2 (FTIR monolayer at 0.1 mm height) and a printing speed of 30 mm/s and 20 mm/s for XY and XZ specimens, respectively, according to ISO/ASTM 52,921 [45]. The printing temperature for all the specimens was set to 235 °C for neat ASA and 260 °C for the composites. The platform temperature was 100 °C in all cases.

Table 2 summarizes the composition of the ASA-BF composites, the surface treatments applied, and the manufacturing processes used with the different composites developed in this research.

### 2.4. Characterization

Mechanical characterization was carried out in injected and printed specimens in a Shimadzu AGS-X (Shimadzu Europa GmbH, Duisburg, Germany) using a constant speed of 1 mm/min, according to ISO 527.

The fibers and fracture surfaces of the tensile testing specimens were observed in a SEM Nova NanoSEM 450 (FEI, Hillsboro, OR, USA) operated at 1.50 kV. The samples for SEM were previously sputtered with a few nm layers of Au in a Balzers SCD 004 Sputter Coater (Oerlikon Balzers, Schaumburg, IL, USA).

A Bruker Alpha spectrometer (Bruker, Billerica, MA, USA) was used for the Fourier-transformed infrared (FTIR) spectroscopy analysis. The spectra were taken by attenuated total reflectance (ATR) mode in the range of 4000–650 cm^−1^, with a spectral resolution of 4 cm^−1^. All the measurements were repeated at least 3 times.

The CTE was measured by thermomechanical analysis (TMA) in a TMA PT1000 dilatometer (Linseis, Robbinsville, NJ, USA), measuring the variation in the length of the specimens by applying a heating rate of 5 °C/min from 20 °C to 80 °C. The measurements were repeated 3 times in order to ensure the reproducibility of the results.

## 3. Results

### 3.1. Influence of the Surface Modification of BF in the Mechanical Properties of the Composites

First, different surface treatments were applied to the BFs in order to achieve the best compatibility with the ASA matrix. For this purpose, a fixed amount of 5 wt% BF was used in all cases. The samples were prepared by injection molding. This was done as a first step to identify the best mechanical performance of the ASA-BF composites manufacturing the materials via FFF. A control in absence of BF was also done to compare the mechanical properties. Figure 2 shows the characteristic strain-stress curves of the different specimens. The characteristic mechanical parameters (Young’s modulus, tensile strength and elongation at break) dissected from these curves are presented in Figure 3.

As it can be observed in Figure 2, all the composites show an increase in their Young modulus when compared to neat ASA. These results evidence that, as expected, a content of only 5 wt% BF leads to more rigid materials, in agreement with previous findings [15,46]. In particular, all treatments led to an increase of approximately 10% in the Young modulus, except for the fibers treated in acidic conditions (BF-a), where the increase was around 5–6%. Only composites prepared with BF-s exhibited an increase in the maximum tensile strength when tested, reaching values above 44 MPa. The rest of the composites showed lower mechanical strength. This behavior indicates that BF-c, BF-a and BF-b did not promote an enhancement in the adhesion between the BF with the ASA matrix.

Finally, the elongation at break decreased for all the BF composites, except for BF-b. However, these differences are not really significant due to the large error associated to these results. The only significant difference in this case is the embrittlement of the BF-a composites, which is well in agreement with the decrease in the other mechanical properties. However, interestingly, the rest of the composites exhibit a plastic deformation similar to that of pure ASA, as it can observed in Figure 4. In the case of BF-a, some white lines perpendicular to the axial deformation can be appreciated along the samples. This seems to indicate an inhomogeneous plasticization, suggesting that the acidic treatment led to an embrittlement of the BF, making it weaker than the BF-c. The rest of BF composites present a continuous reduction of their section, evidencing a plastic behavior. This plastic behavior was not expected, since usually the incorporation of fibers leads to a more drastic ductile to fragile transition [43,47], suggesting a good integration of the fibers in the ASA matrix in these cases.

To gain further insight into the effect of the different treatments applied to the BFs, SEM analysis was done to the different BFs, showed in Figure 5. BF-c and BF-b showed a smooth surface, which seems to indicate that these treatments were not aggressive enough to damage the structure of the BFs. The BF-b treatment did not lead to any significant modification in the BFs, since the mechanical parameters dissected for BF-b are not significantly different to those obtained for BF-c. In the case of BF-a, it shows small cracks in the surface even before compounding the composite. Consequently, these cracks are likely to act as nucleation points of crazes inside the composite during the tension test [37]. This explains the weaker behavior of the BF-a composites observed in Figure 2 and Figure 3, compared to all other composites. SEM image of BF-s show enhanced roughness in the surface and shows some regions where traces of the silane coupling agent can be found. This increases the surface roughness of the fiber, while it also modifies the wettability of the BFs, increasing their hydrophobicity. The combination of these two factors causes an increase in the adhesion between the interface of the BFs with the polymeric matrix, caused by the formation of Si-O covalent bond [35,48]. These results are also in agreement with previous results from other authors, where different fibers were silanized before compounding [15,31,32,33,34].

In conclusion, the results obtained by the mechanical testing together with the SEM analysis of the BFs after different treatments evidence that the surface modification of the BF with the silane coupling agent leads to a better performance as a reinforcing agent in polymer-based composites. For this reason, from here on, the research is only focused on the BF-s composites.

### 3.2. ASA-BF Composites Prepared by FFF

After the study of the different treatments for the BF, filaments of ASA and ASA containing 5 and 10 wt% BF-s were manufactured for FFF. Then, standardized specimens for tensile test (in XY and XZ orientation) were printed as indicated in the materials and methods section. In a similar way to what was done with the composites prepared by injection molding, Table 3 shows the mechanical parameters dissected from the tensile test curves for each of the materials printed in this study.

In general, the mechanical properties of the XY specimens show higher values than those of the XZ specimens. This anisotropy is characteristic of these AM processes [49,50,51]. The addition of fibers complicate the entanglement of the polymer chain in the interlayer region to some extent, hindering the adhesion between the layers. This explains the lower mechanical properties obtained in XZ orientation for composites compared with neat ASA [43].

The XY composites exhibit a higher Young modulus values than neat ASA XY, in a similar way to what was previously observed in Figure 3 for the composites prepared by injection molding, when the incorporation of BF-s increases the stiffness and strength compared with neat ASA. In this case, an increase from 5 wt% to 10 wt% of BF-s does not enhance the mechanical properties but maintains the ductility of the composite compared to ASA.

Then, the fracture surfaces of the BF composites were examined by SEM (Figure 6), where the distribution of BF-s within the polymer matrix can be appreciated after tensile test. The fibers act as a proper reinforcement, increasing the tensile strength but might act as well as potential crack initiation points that lead to the previously observed embrittlement of the material. In addition, the porosity observed can be attributed to the presence of air bubbles that may remain entrapped during the compounding and by the different CTEs of the phases involved (polymer and fiber), which create a localized porosity preferably at the end of the fibers [52].

Infrared spectroscopy analysis of the composites by FTIR in ATR mode was done to identify any supramolecular interactions between the ASA matrix and the BF. Figure 7 shows the spectra of the neat ASA (black), together with those of ASA + 5BF-s (red) and ASA + 10BF-s (blue). For comparative purposes, all the spectra were normalized to the C≡N stretching band at 2237 cm^−1^. Some differences in the stretching of the -CH_2_ and -CH_3_ bands in the range of 2800–300 cm^−1^, associated with the aliphatic chains of ASA and to a lesser extent to the hydrophobic moieties of the APTES immobilized on to the BF-s, were observed. These differences suggest that there are some interactions taking place with the BF-s. These results, together with the microscopy analysis presented in Figure 6, support that in this case the silanization of the BF has led to a significant increase in the compatibility with the ASA matrix.

Thermal expansion coefficient of the BF-s composites was dissected from TMA analysis. Figure 8 shows the dimensional change (ΔL) of the specimens as a function of the temperature for ASA and 5–10 wt% BF-s composites, manufactured in two different printing directions. When the BF-s content is increased, a significant decrease in the ΔL of the specimens tested is observed for both XY and XZ specimens. This is caused by the nature of the BF, having an intrinsic CTE close to zero. When the two printing orientations (i.e., XY, XZ) are compared, a clear anisotropy is observed, with ΔLvalues 60% lower for XY specimens.

The CTE values, obtained from the TMA curves, are summarized in Table 4. It can be observed that the printing direction has a clear influence in the CTE, and 10 wt% BF-s ASA composites decrease the CTE down to a 65–75% when compared to neat ASA, for both XY and XZ printing directions. The anisotropy intrinsic in the FFF manufacturing process is also evidenced by CTE analysis, leading to worse reinforcement properties when the materials are printed vertically, due to the deficient adhesion between layers during the deposition of the molten polymeric layers [53]. In a similar manner, this also causes that ASA and its composites have a higher tendency to increase its size, preferably in this direction. In the XZ configuration, the BFs are not so interconnected to each other, so they are not able to prevent in the same amount the deformation in this direction than in XY.

The incorporation of fibers restrains the mobility of the polymer chains [43,54], justifying the reduction CTE in the composites. In the XY specimens, the fibers are preferentially oriented along the printed bead [7], accounting for the greater CTE reduction in this direction than XZ, where fibers contribute to a lesser extent to these mechanisms. To the best of our knowledge, this is the first investigation of 3D-printed ASA composites reinforced with BF, but other authors have also reported the influence of the printing direction on the CTE using other polymer matrixes and reinforcements. Arif et al. [55] studied the influence of PEEK composites reinforced with carbon nanotubes and graphene nanoplatelets via FFF, with a clear anisotropy between the two printing orientations. This anisotropy is affected not only by the printing orientation but also by the raster angle of the layers [56]. More interestingly, Hoskins et al. [57] reported the influence of 20 wt% CF on the CTE using ABS as a polymeric matrix. In their case, they observed a difference of practically one order of magnitude, from 19.4 µm/m °C to 128 µm/m °C in XY and XZ orientation. Similar results were also reported by Billah et al. [12] for ABS reinforced with CF and GF, reaching CTE values down to 22 and 43 µm/m °C, respectively. In our case, we show that with smaller amounts of BF (5–10 wt%), the CTE, particularly in the XZ direction, decreases substantially, below 100 µm/m °C. Moreover, in our case, the difference in the CTE values between XY and XZ is smaller, indicating that the anisotropy of these materials is smaller.

Reducing the CTE has a significant impact on the performance of the printed parts by not only improving dimensional accuracy, but also minimizing their distortion. An illustrative example is presented in Figure 9, where it is shown that the composites printed with BF do not exhibit any warping compared to the pure ASA after cooling, due to their lower CTE. In the part printed with ASA + 10BF-s, some agglomerates are observed locally. This may explain why a further increase in the tensile strength was not observed from 5 to 10 wt% FB-s, implying that 10 wt% BF may lead to local defects when printing.

These results match previous studies carried out by the authors [58] and others, where carbon fiber is used to reduce the warping of printed parts [59,60]. Hence, BFs are a potential alternative to CF to reduce the distortion of parts manufactured by FFF.

## 4. Conclusions

In this work, we have evidenced the potential of ASA-BF composite materials for additive manufacturing as an alternative to CF and GF. Different surface treatments for the BFs have been studied, obtaining the best performance for silanized BFs with APTES (BF-s). On the other hand, the BFs treated in acidic conditions led to very brittle and weak composites, due to the appearance of cracks in the BF during this treatment. Then, different composites with 5 and 10 wt% BF-s were manufactured by FFF. It was observed that an increase from 5 to 10 wt% BF-s did not lead to a significant enhancement in the mechanical properties, although it was proved that the silanization treatment avoids the embrittlement of the material, as it happens for other composites. This was evidenced by microscopy and spectroscopic analysis of the composites, proving that the BF-s are well embedded in the ASA matrix. Finally, the CTE of the composites was also evaluated. We proved that the BF-s are able to decrease this parameter in different printing directions. This represents a major advantage in FFF, since low CTE values prevent the delamination of the objects and warping issues. Hence, we consider that these materials are a promising alternative to the current composites used in FFF, with a potential implementation in large format additive manufacturing applications, where big parts and components can be designed and printed on demand.

## Figures and Tables

**Figure 1 polymers-14-03216-f001:**
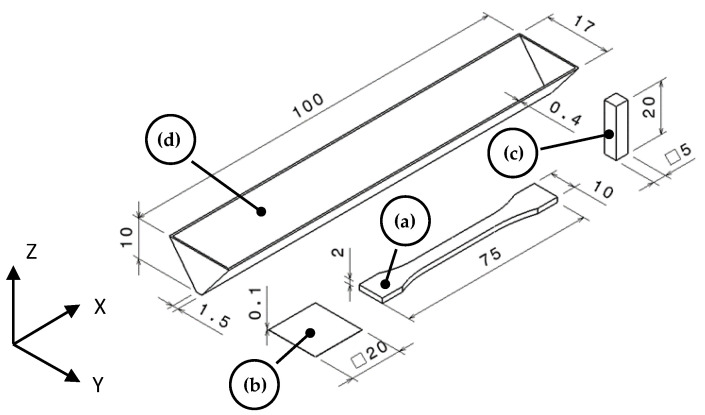
Specimen design to be printed for the characterization in this research, (a) 1BA tensile testing (XY orientation); (b) FTIR monolayer; (c) CTE paralepidid (XZ orientation); (d) warping V hollow specimen. Dimensions in mm.

**Figure 2 polymers-14-03216-f002:**
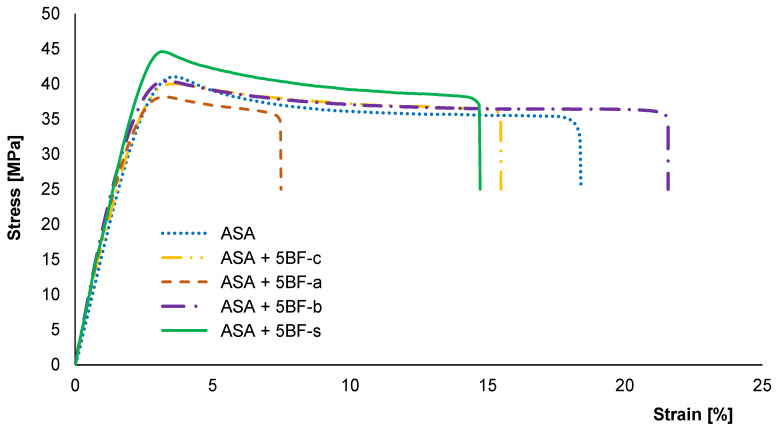
Tensile testing representative stress-strain curves of ASA and composites prepared with 5 wt% of BF after different surface treatments by IM.

**Figure 3 polymers-14-03216-f003:**
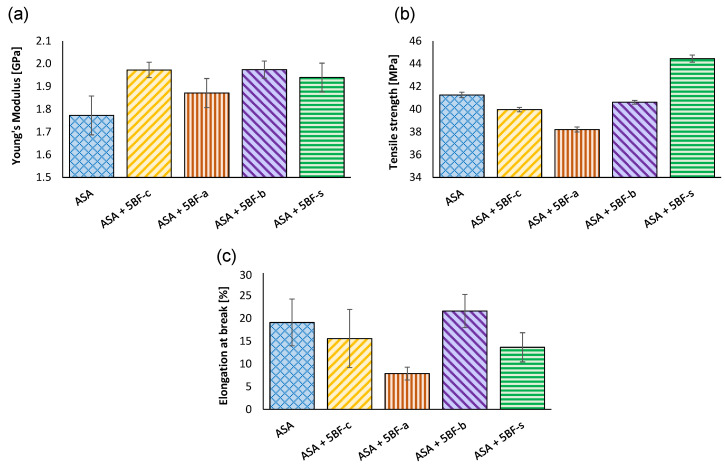
(**a**) Young modulus; (**b**) tensile strength and (**c**) elongation at break of ASA and composites prepared with 5 wt% BF after different surface treatments by IM.

**Figure 4 polymers-14-03216-f004:**
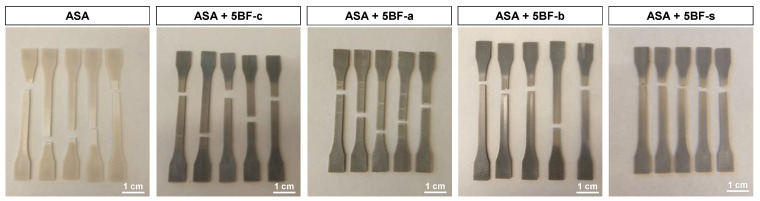
IM samples after the tensile test of ASA and 5 wt% BF composites studied in this work.

**Figure 5 polymers-14-03216-f005:**
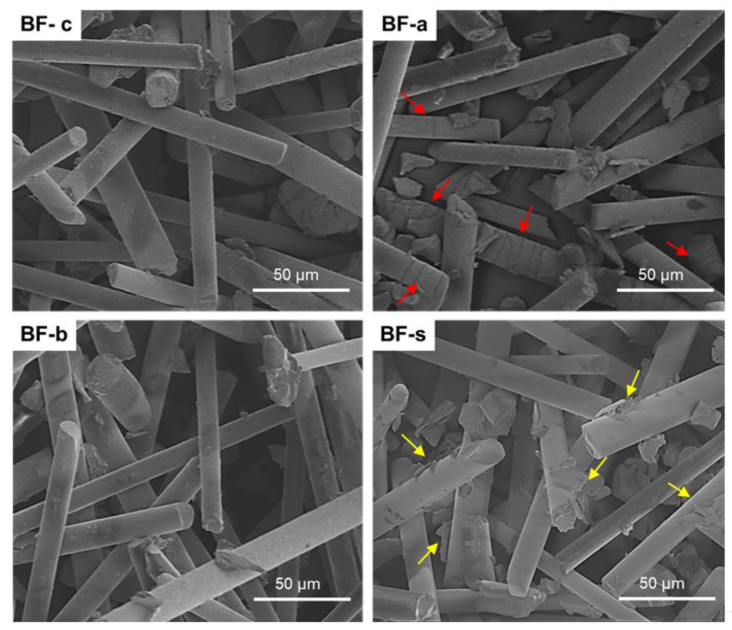
SEM images of the different fibers evaluated. Red arrows in BF-a highlight the cracks in fibers of this composition. Yellow arrows in BF-s highlight roughness in surface of the fibers.

**Figure 6 polymers-14-03216-f006:**
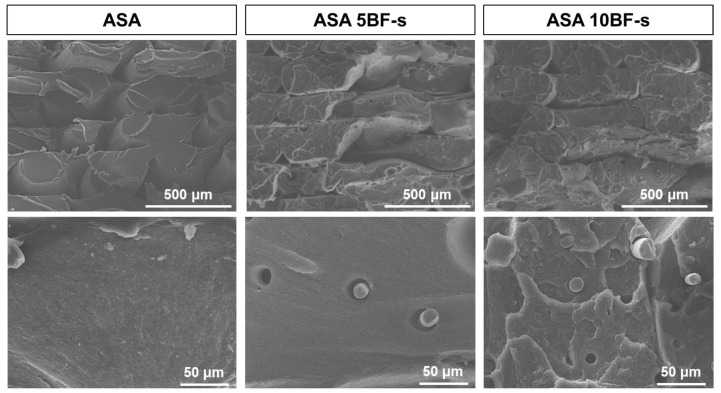
SEM images of fracture surface after tensile testing of ASA and ASA composites manufactured by FFF.

**Figure 7 polymers-14-03216-f007:**
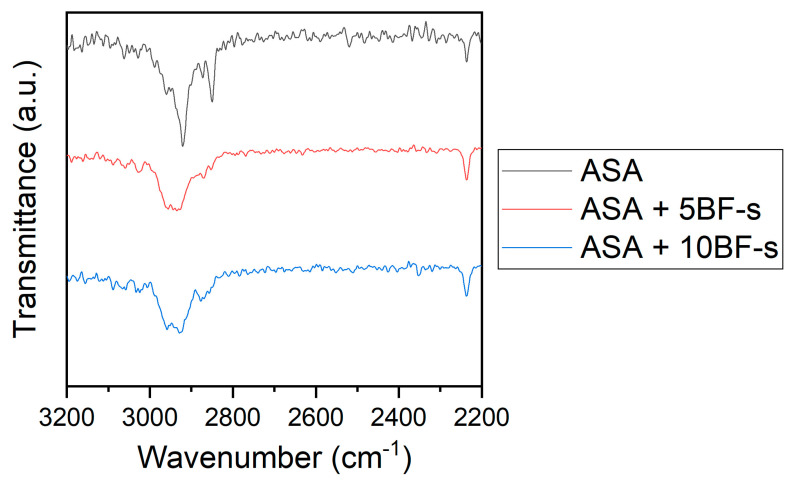
ATR spectra of ASA (black), ASA + 5BF-s (red) and ASA + 10BF-s (blue). Green frame indicates the region of interest.

**Figure 8 polymers-14-03216-f008:**
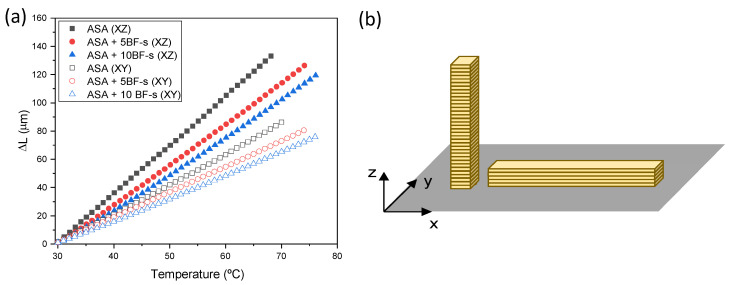
(**a**) TMA curves of ASA (black), ASA + 5BF-s (red) and ASA + 10BF-s (blue). Filled symbols represent the specimens tested vertically (labelled as XZ) and hollow symbols represent the specimens printed horizontally (labelled as XY); (**b**) cartoon depicting the disposition of the specimens in the printing platform for easier interpretation of (**a**).

**Figure 9 polymers-14-03216-f009:**
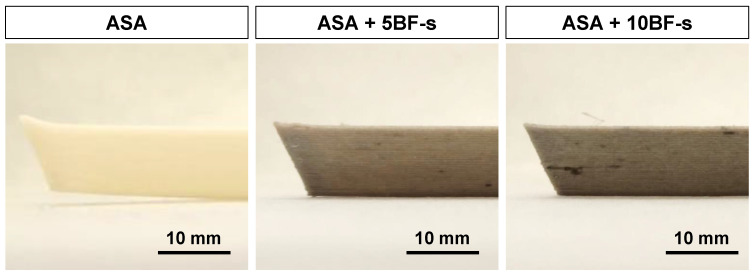
Side view of warping parts printed in ASA, ASA + 5BF-s and ASA + 10BF-s.

**Table 1 polymers-14-03216-t001:** Chemical composition of the BFs used in this research.

wt%	SiO_2_	Al_2_O_3_	Fe_2_O_3_	CaO	MgO	TiO_2_	Na_2_O	Others
Minimum	45	12	5	6	3	0.9	2.5	2.0
Maximum	60	19	15	12	7	2.0	6.0	3.5

**Table 2 polymers-14-03216-t002:** Summary of all the composites with denomination, composition and manufacturing processes used in this work.

Denomination	Composition	BF Treatment	Manufacturing Process
ASA	100 wt.% ASA	-	IM/FFF
ASA + 5BF-c	95 wt.% ASA + 5 wt.% BF-c	Calcined	IM
ASA + 5BF-a	95 wt.% ASA + 5 wt.% BF-a	Acid (H_2_SO_4_)	IM
ASA + 5BF-b	95 wt.% ASA + 5 wt.% BF-b	Basic (NaOH)	IM
ASA + 5BF-s	95 wt.% ASA + 5 wt.% BF-s	Silanization (APTES)	IM/FFF
ASA + 10BF-s	90 wt.% ASA + 10 wt.% BF-s	Silanization (APTES)	FFF

**Table 3 polymers-14-03216-t003:** Mechanical properties of specimen printed in XY and XZ orientation by FFF of ASA, ASA + 5BF-s and ASA + 10BF-s.

Denomination	Young Modulus [GPa]		Tensile Strength [MPa]		Elongation at Break [%]	
	XY	XZ	XY	XZ	XY	XZ
ASA	1.6 ± 0.1	1.03 ± 0.11	41.3 ± 0.2	17 ± 1	3.88 ± 0.16	1.9 ± 0.1
ASA + 5BF-s	1.8 ± 0.1	1.2 ± 0.1	42 ± 2	11 ± 1	4.96 ± 1.58	1.1 ± 0.2
ASA + 10BF-s	1.8 ± 0.1	1.01 ± 0.05	42.1 ± 0.6	4 ± 1	3.96 ± 0.30	0.5 ± 0.1

**Table 4 polymers-14-03216-t004:** CTE of ASA, ASA + 5BF-s and ASA + 10BF-s in the linear region of 40–70 °C printed in XY and XZ orientation.

Denomination	CTE [µm/m °C]	
	XY	XZ
ASA	69.3 ± 16.8	115.4 ± 24.3
ASA + 5BF-s	60.9 ± 13.1	91.2 ± 21.6
ASA + 10BF-s	52.1 ± 12.6	79.9 ± 20.8

## Data Availability

Not applicable.

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
