# Peer review of "Basalt Fiber Composites with Reduced Thermal Expansion for Additive Manufacturing"

_polymers, 2022, doi:10.3390/polym14153216_

Round 1
Reviewer 1 Report
This paper studies the basalt fibers reinforced ASA, as a novel topic in the field. The paper is well prepared and with good contributions. Minor comments, listed as follows, should be addressed before publication.
For the tensile tests, please add failure mode descriptions with some images.
Please re-organize well for the tables, especially the second table. It should be ordered as Table 2, but the table title is missed.
The review and introduction for the thermal performance of BF polymer can be added and extended. Please refer to “Degradation of the in-plane shear modulus of structural BFRP laminates due to high temperature”, and “Effects of elevated temperatures on the mechanical properties of basalt fibers and BFRP plates”.
Author Response
- This paper studies the basalt fibers reinforced ASA, as a novel topic in the field. The paper is well prepared and with good contributions. Minor comments, listed as follows, should be addressed before publication.
We really appreciate the overall positive opinion on the submitted work and the constructive comments about it. Please, find our answers below and highlighted in yellow in the manuscript.
- For the tensile tests, please add failure mode descriptions with some images.
We thank the reviewer for pointing this fact. A figure with samples after the tensile tests has been included in the revised version (Figure 4), and we have enlarged the discussion of the fracture of the specimens.
- Please re-organize well for the tables, especially the second table. It should be ordered as Table 2, but the table title is missed.
We thank the reviewer for catching this typos. We have carefully inspected the tables and include the title in Table 2. Also, in Table 2 we have included a new column indicating the denomination and the composition of each material studied.
- The review and introduction for the thermal performance of BF polymer can be added and extended. Please refer to “Degradation of the in-plane shear modulus of structural BFRP laminates due to high temperature”, and “Effects of elevated temperatures on the mechanical properties of basalt fibers and BFRP plates”.
We acknowledge the suggestion. We have extended the introduction section including the references indicated.
Reviewer 2 Report
In this paper, the authors investigated a series of composites based on ASA reinforced with basalt fibers for fused deposition modeling. Main comments were listed as follows:
1. Why the authors chose ASA as the polymer matrix and BF as the additive? What’re the unique advantages this composite system can provide in comparison with other composites such as ASA/CF?
2. What’s the title of the table between pages 3 and 4? Also, it’s not well-organized.
3. a), b), c)…were not marked in Fig. 1.
4. Detail company information needed to be provided in Materials and Methods. For example, the mechanical tester.
5. Figure 2, please add “stress” to y-axis and “strain” to x-axis. What’s εR? Use different line types to show different materials. Otherwise, it’s difficult to distinguish in black and white
6. Figure 3, please add different infill patterns to show the mechanical properties of different materials.
7. More discussions needed to be provided in Chapter 3. For example, why the smaller amounts of BF decreased the CTE in the xz direction? The authors cited many published results. But the materials used in these published papers were different from the authors’ material. How to compare or borrow the explanation if the materials were different?
8. In 3D printing, the rheological properties of build materials were also very important. How did the addition of BF affect the rheological properties, e.g., viscosity, shear moduli, solidification temperature, etc.?
Author Response
In this paper, the authors investigated a series of composites based on ASA reinforced with basalt fibers for fused deposition modeling. Main comments were listed as follows:
We really appreciate the comments done by the reviewer. Please, find our answers below and highlighted in yellow in the manuscript.
- Why the authors chose ASA as the polymer matrix and BF as the additive? What’re the unique advantages this composite system can provide in comparison with other composites such as ASA/CF?
We really appreciate the reviewer's appreciation for the decision to evaluate ASA BF in additive manufacturing. To the best of our knowledge, no research has been done combining ASA and BF, so it would be the first study to be published on this topic. In previous works we evaluated the incorporation of CF in ASA, however the high price of CF and its availability limits in a certain way the use of ASA CF in industrial applications. The use of BF is a feasible alternative as a reinforcement with a significant price reduction, compared to CF. We include reference 18, which includes an economic comparison between different fibers. In addition, ASA has excellent resistance to UV light, opening the door to printing parts with excellent performance for outdoor applications. The good results obtained in the reduction of the CTE and warping test would make it possible to obtain parts without distortions, which would even allow the use of this material in large format printers.
[18] A. Saleem, L. Medina, M. Skrifvars, and L. Berglin, “Hybrid Polymer Composites of Bio-Based Bast Fibers with Glass, Carbon and Basalt Fibers for Automotive Applications-A Review,” Molecules (Basel, Switzerland), vol. 25, no. 21. NLM (Medline), Oct. 25, 2020. doi: 10.3390/molecules25214933
- What’s the title of the table between pages 3 and 4? Also, it’s not well-organized.
We have included the title of Table 2 and also completed the table with a new column to independently indicate the denomination and the composition of each material studied.
- a), b), c)…were not marked in Fig. 1.
We acknowledge the review’s suggestion, and we update the figure indicating the marks a), b), c) and d) in it.
- Detail company information needed to be provided in Materials and Methods. For example, the mechanical tester.
We have carefully reviewed the Material and Methods section, including all companies in devices and materials.
- Figure 2, please add “stress” to y-axis and “strain” to x-axis. What’s εR? Use different line types to show different materials. Otherwise, it’s difficult to distinguish in black and white
We restyle the Figure 2 with different type of lines and including “stress” and “strain” in the axis. We consider that with the new configuration is clearer in black and white version.
- Figure 3, please add different infill patterns to show the mechanical properties of different materials.
Following the previous recommendation, we have restyled the Figure 3 with different patterns for a clearer interpretation in black and white version.
- More discussions needed to be provided in Chapter 3. For example, why the smaller amounts of BF decreased the CTE in the xz direction? The authors cited many published results. But the materials used in these published papers were different from the authors’ material. How to compare or borrow the explanation if the materials were different?
We discussed more in detail the reduction of CTE with the incorporation of BF in ASA and the comparison with previous works. In general, loading a thermoplastic with an inorganic filler (i.e. CF, GF, graphite nanoplates, inorganic particles…) is expected to decrease the CTE to a certain extent, since they prevent the mobility of the polymer chains. We added the following paragraph:
‘The incorporation of fibers restrains the mobility of the polymer chains [45], [56], justifying the reduction CTE in the composites. In the XY specimens, the fibers are preferentially oriented along the printed bead [9], accounting for the greater CTE re-duction in this direction than XZ, where fibers contribute to less extent to these mech-anisms. To the best of our knowledge, this is the first investigation of 3D-printed ASA composites reinforced with BF, but other authors have also reported the influence of the printing direction on the CTE using other polymer matrixes and reinforcements.’
[9] H. L. Tekinalp et al., “Highly oriented carbon fiber–polymer composites via additive manufacturing,” Com-posites Science and Technology, vol. 105, pp. 144–150, 2014, doi: 10.1016/j.compscitech.2014.10.009.
- In 3D printing, the rheological properties of build materials were also very important. How did the addition of BF affect the rheological properties, e.g., viscosity, shear moduli, solidification temperature, etc.?
We deeply appreciate the reviewer's suggestion and agree with the importance of rheological properties of materials in 3D printing. Actually, we measured the melt flow rate (MFR) of the printed materials, but the results did not show significant differences between neat ASA and the BF composites due to the relatively high errors obtained in the measurements. For this reason, we do not feel confident enough including this information.
In any case, we would like to highlight that in this presented work we just would like to stress the suitability of BF-reinforced ASA as an alternative to other composites for FFF, opening the door to further research such as the one proposed by the reviewer, which would be outside the scope of this work. We plan to cover a more in deep rheological study as part of a future work, focused in large format FFF applications.
Round 2
Reviewer 2 Report
The authors answered all my questions. I think the paper is ready to go.
This manuscript is a resubmission of an earlier submission. The following is a list of the peer review reports and author responses from that submission.
Round 1
Reviewer 1 Report
The article focused on the effects of fiber surface treatment on the mechanical properties and CTE of Basalt fiber/ASA composites. The potential application of the composite as 3D printing raw material is a timely topic, but the work is not of sufficient novelty to be published, and the study suffers major flaws in its questionable methodology, poor quality of figures, and lack of in-depth analysis.
- Only traditional surface treatment techniques and agents were used in the work and obviously not optimized. The results are quite predictable with common sense.
- The property variations of the IM samples were not further investigated by considering other key factors like fiber dispersion, orientation, true L/D ratio after multiple cycles of processing, etc.
- The FFF samples didn’t perform much improvement with the “best” silane treatment fibers as compared to that with calcined fibers, which was not explained.
- Regarding the decrease of properties with higher load of fibers, some speculations were given but not verified. Now the question is: What are the true percentage of fibers in the composite samples? Since all the conclusions were made based on the percentage of the fibers, understanding the facts and the reasons is critical not only for the high fiber loading samples, but also for the low fiber loading samples. This is equally essential for comparing samples with different fiber treatments since the treatment would affect the dispersion of the fibers and the rheological behavior of the melts, and then affect the flowing/clogging at the nozzle.
- If the clogging of fibers at nozzle is confirmed as proposed, it would have affected all the samples rather than the high fiber loading ones only, which means that wrong equipment was chosen for the experiment and all the data presented in the paper is questionable. Therefore, the whole work should be repeated with an appropriate equipment.
- FTIR-ATR was used to study the supramolecular interactions between the ASA matrix and the BF. It is questionable that this method can collect enough signal to serve the purpose of current work since only sample surface were scanned while fiber content is quite low (as shown in Figure 5) and most of the fibers are embedded underneath.
- The quality of the SEM pictures in Figure 3 and the OM pictures in Figure 5 is quite poor.
Reviewer 2 Report
In order to enhance the quality and strengthen this work, authors are advised to address some issues.
Page 3, lines 107-108, authors described preparation of the samples by single screw extruder. Like emphasized, they utilized single-screw extruder two times in order “to obtain a more homogeneous composite”. Since the polymers are prone to thermal degradation, even at preparation stage, the question is: why authors didn’t use twin-screw extruder? Thanks to their high mixing ability, twin-screw extruders are usually utilized in compounding and reactive polymer processes. On the other hand, upon their relatively poor mixing ability, single-screw extruders are not considered as the best solution to produce composites.
Page 4, lines 140-143 authors described FT-IR analysis. What did you mean by “All the measurements were repeated at least 3 times.”
Page 6, lines 174-176, authors mentioned some large error: “However, these differences are not really significant due to the large error associated to these results.” Please explain.
Page 8, lines 243-245, authors suggested to use bigger nozzle, rather than keep decreasing the size of the fibers, which would enhance the mechanical properties of the composite to a lesser extent. In my opinion it is better to use smaller size particles to enhance mechanical and other properties of polymer composites. If the size is smaller, the specific surface of particles or fibers is larger. The same page, lines 270-271 authors stated “However, some differences in the stretching of the -CH2 and -CH3 bands in the range of 2800-300 cm-1”. Did authors mean 2800-3000 cm-1?
Since in this work polymer composite is subjected to thermal processing (two times extrusion + injection/printing), the major drawback of this work is that authors didn’t evaluate the effect of thermal processing on the composite during the preparation stage. Specifically, to evaluate if the thermal degradation of the samples occurred. Differential scanning calorimetry (DSC) and thermogravimetric analysis (TG) could be utilized. DSC would provide info about melting and glass transition, while TG would give useful info about thermal stability and degradation that may occur during sample preparation; degradation could trigger changes in the structure and consequently in mechanical properties. These analyses are logical step that serve as a check point, like FT-IR is used in this work, which would make this work stronger and complete.